# Analysis of Orientation Errors in Triaxial Fluxgate Sensors and Research on Their Calibration Methods

Xiujuan Hu[1,2], Shaopeng He[1,2], Qin Tian[1,2], Alimjian Mamatemin[3] , Pengkun Guo [1,2], Guoping Chang [1,2]

[1] Earthquake Administration of Hebei Province, Shijiazhuang, 230071, China
[2] National Field Scientific Observation and Research Station for Huge Thick Sediments and Seismic, Xingtai, 054000, China
[3] Earthquake Administration of Xinjiang Province,Kashi, 843300, China

*Correspondence to*: Xiujuan Hu (huxiujuan1260@163.com)

**Abstract.** Three-axis magnetic flux gate sensors are widely used in Chinese geomagnetic observatories, but due to their directional errors, it is necessary to study error correction methods to improve measurement accuracy. Firstly, the mechanism of directional errors produced by three-axis magnetic flux gate sensors is analyzed, followed by the development of measurement tools for conducting directional error measurement experiments on the high-precision three-axis magnetic flux gate sensors of the Chinese FGM-01 series. Experimental results show that correcting the Z-axis and D-axis directional errors is essential. The observation data after error correction, whether in terms of the standard deviation of its all-day baseline values or the relative difference magnitude with the reference instrument, significantly decrease, demonstrating the clear correction effect and proving the effectiveness of this correction method.

Keywords:Tri-axis Magnetic Flux Gate Sensor, Orientation Error, Calibration

## 1 Introduction

Three-axis fluxgate sensors have the advantages of high resolution, low power consumption, and low cost, and are widely used in measuring the geomagnetic field signal (Langel et al.,1988; Tohyama et al.,1988a,1988b; Ejiri et al.,1988;Crassidis and Lai, 2005). Currently, nearly 200 sets of three-axis fluxgate magnetometers, mainly GM4 type (Figure 1a), GM4-XL type, and FGM-01 type (Figure 1b), are installed in the Chinese geomagnetic observatories. Most observatories install two or more sets of such instruments for parallel observations, aiming to ensure the continuity and integrity of the observation data and to facilitate timely detection and identification of potential issues in the data. The ideal measurement value of a three-axis fluxgate sensor should be equal to the true value of the measured geomagnetic field variation(Luo et al.,2019;Wu,2008). However, due to limitations in manufacturing and installation processes, errors such as non-orthogonality, zero offset, and temperature drift exist in three-axis fluxgate sensors unavoidably(Včelák et al.,2006; Foster and Elkaim, 2008; Pang, 2011). Studies have shown that these errors can lead to deviations of the sensor's measurement values from the true values of the measured geomagnetic field, significantly affecting its measurement accuracy. Therefore, it is of great significance to correct the errors of the sensor(Zhu et al.,2005;Li 2008).

Research on the error of three-axis magnetic fluxgate sensors in the past has typically only considered the systematic error of the sensors(Liu et al., 2022), with relatively little study on the directional errors introduced during sensor installation.Wang Xiaomei et al.(2017) analyzed the variation patterns between the orientation of the instrument, the level of the base, and the observed data of each component of the geomagnetic field based on theoretical calculations and station experiments,

providing quantitative relationships. Liu Cheng et al.(2019) established a three-axis calibration model
for the magnetic fluxgate magnetometer, determined the attitude angles and scale factor coefficients of
the instrument, and then corrected the actual observation data based on the calculation results. The
author once developed a measuring device in 2016 to measure and correct the directional errors of the
D magnetic axis, eliminating the daily variation distortion recorded by the magnetic fluxgate
magnetometer at Hongshan Observatory. However, the aforementioned algorithms or measurements
are somewhat difficult (Zhu,2004) and not easy to implement, or only focus on the method of
correcting the directional errors of the sensor's D magnetic axis, without yet conducting research on
correcting the directional errors of the Z magnetic axis, all of which have shortcomings that need
improvement.
This study analyzes the mechanism of directional error generation of the three-axis magnetic flux gate
sensor, measures the directional error of the sensor using homemade measurement tools, and corrects
the measurement results when reorienting. Finally, by comparing the changes in the actual
measurement data before and after correction, the correction effect is analyzed.

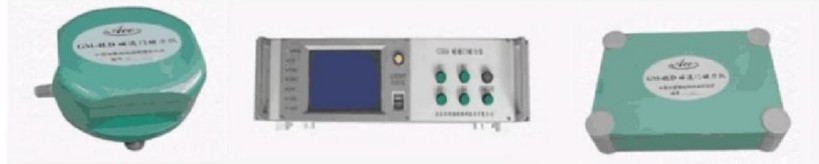

**(a) GM4 Type Fluxgate Magnetomete**

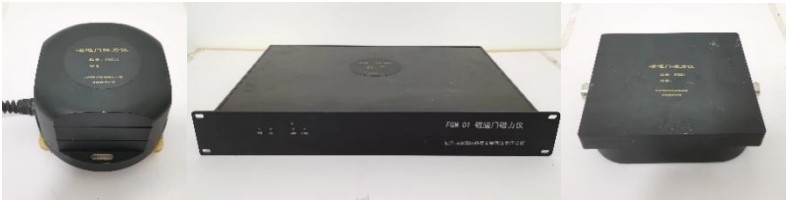

**(b) FGM-01 Type Fluxgate Magnetometer**

**Figure 1: Fluxgate Magnetometer**

**2 Analysis of Directional Errors in Three-Axis Fluxgate Magnetometer Sensors**

This article mainly analyzes the directional errors generated during the installation of a three-axis
fluxgate magnetometer, ignoring the errors of the three-axis orthogonality. Therefore, during the
experiment, it is assumed that the three-axis of the fluxgate sensor is in a perfectly orthogonal ideal
state. When installing the fluxgate instrument, a sensor that measures the declination angle D is usually
used for orientation. Currently, Chinese geomagnetic observatories typically orient sensors in the
following manner (referred to as the traditional orientation method). The first step is to select a day
with calm magnetic fields, adjust the base angle screws of the sensor to center two mutually
perpendicular bubbles, thereby determining the orientation of the Z-magnetic axis. The second step is
to rotate the sensor horizontally to control the output value of the magnetic declination angle D within
the range of -50-50nT, thus determining the orientation of the D-magnetic axis. The orientation of the
H-magnetic axis is determined as the D-magnetic axis is determined.
Assuming that the horizontal plane HOD of the three-axis fluxgate sensor is absolutely horizontal with
the ideal coordinate system XOY, the angle of rotation of the sensor in the HOD plane with the H
element as the axis is called the orientation error angle α (Wang et al., 2017). As shown in Figure 2,
due to the presence of the orientation error angle α, the measurement values of the two elements in the
horizontal plane mutually include each other's components. That is, the measurement value of the D
element is the sum of the projections of the real D element and the H element in its measurement
direction, which is expressed as
$D_1 = D'_0 - H''_0 = D_0 \cos\alpha - H_0 \sin\alpha$
Similarly, the measurement value of the H element is
$H_1 = H'_0 + D''_0 = H_0 \cos\alpha + D_0 \sin\alpha$
where $D_0$ and $H_0$ are the values of the magnetic field D and H elements in the ideal coordinate system,
$D'_0$ and $D''_0$ are the projections of the D element in the HOD plane when there is an orientation error
angle $\alpha$, and $H'_0$ and $H''_0$ are the projections of the H element in the HOD plane when there is an
orientation error angle $\alpha$. Since the value of the D element and $\alpha$ are relatively small, $D_0 \sin\alpha$ can be
omitted. Therefore, if there is an orientation error angle $\alpha$, it has a greater impact on the recorded data
of the D element.

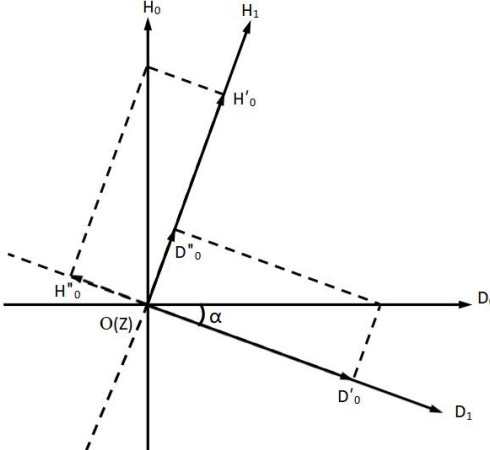

**Figure 2 introduces the coordinate reference system of the magnetic sensor with the directional error**
**angle α.**
In traditional orientation methods, it is believed that controlling the output value of the magnetic
declination D within the range of -50 to 50 nT results in a relatively small orientation error angle.
However, due to some ferromagnetic substances inherent in the triaxial fluxgate sensor being
magnetized by the environmental magnetic field, residual magnetism is generated. This, combined with
zero drift produced by the sensor and data acquisition module, collectively superimposes a fixed
magnetic field on each axis of the fluxgate sensor, causing the measured magnetic field component
values to shift (Luo et al., 2019). Therefore, when the magnetic declination D is oriented, the output
value simultaneously includes the offset of the D magnetic axis, which may result in an increase in the
orientation error angle $\alpha$, leading to inaccurate orientation of the D magnetic axis.
Assume the offset of the D magnetic axis is $S_0$, and the projection value of the magnetic field H on the
D magnetic axis is S, as shown in Figure 3(a), where A represents the magnetic east direction, B
represents the magnetic north direction, and C represents the position of the magnetic axis when the
offset $S_0$ exists and the output value of D is zero. At this point, the output value of the D component is
$S_0 - S$; as shown in Figure 3(b), rotate the position of the D magnetic axis horizontally by 180°, and the
output value of the D element is $S_0 + S$; the numerical value of the offset $S_0$ can be obtained through
calculation.

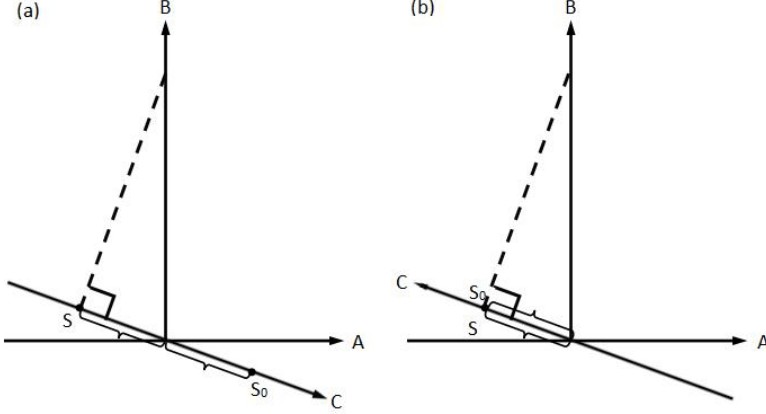


Figure 3: Calculation schematic diagram of offset $S_0$ at the D magnetic axis orientation position (a) and its position after rotating 180°(b)

Furthermore, the angle error caused by the non-horizontal placement of the triaxial fluxgate magnetometer will also lead to mutual influences among the components. Therefore, whether the leveling bubble of the instrument base can ensure that the Z magnetic axis is vertical is also crucial for accurate orientation.

In summary, when installing and orienting the instrument, to ensure accurate orientation, in addition to considering the magnitude of the D output value during orientation, it is also necessary to consider the offset of the D magnetic axis, and at the same time, ensure whether the Z magnetic axis is truly in a vertical state.

## 3 Experiment Introduction

To complete this experiment, a set of non-magnetic rotary platform (hereinafter referred to as the platform) was specifically designed. This platform mainly consists of a weak magnetic plate and a non-magnetic theodolite. The weak magnetic aluminum plate is installed on the non-magnetic theodolite telescope (Figure 4a), and it enables the platform to rotate on a horizontal plane by adjusting the vertical dial and horizontal dial of the theodolite. During the experiment, first, adjust the level of the theodolite, then adjust the vertical dial to make the platform horizontal, then place the sensor on the platform, and check the verticality of the Z magnetic axis by rotating the horizontal dial and observing the output value of the Z magnetic axis. The offset of the D magnetic axis can be measured by adjusting the theodolite horizontal dial.

Before the experiment, two adjustable spirit levels with an accuracy of 10s, perpendicular to each other, were fixed on the top of the sensor with rosin, one of which passes through the magnetic axis (Figure 4b). When the Z axis reaches the ideal vertical state, adjust the spirit level to a horizontal state so that when the sensor is installed in a new location, it can be placed in the same vertical position (Jankowski J and Sucksdoff C, 1996).

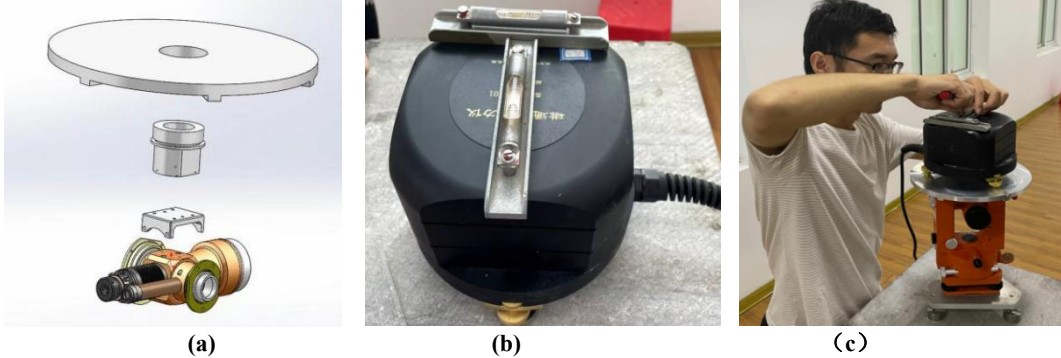

|          (a)          |          (b)          |          (c)          |

Figure 4: Schematic assembly and actual operation of non-magnetic rotation platform

(a) Non-magnetic rotation platform connection schematic diagram  (b) Sensor top with external level  (c) Actual operation of non-magnetic rotation platform

The measurement part of this experiment was conducted in the absolute  measurement room of Hongshan observatory. The FGM-01 magnetic fluxgate magnetometer was used as the instrument under test (see Fig. 4c).

First, we examined the perpendicularity of the Z magnetic axis using the platform. After leveling the non-magnetic theodolite, we began adjusting the sensor's base angle screws to center two mutually perpendicular bubbles. By rotating the platform, we recorded the output values of the Z element at four positions 90° apart, as shown in Table 1. From the first set of data, it can be observed that the output values of the Z element at the four positions are not equal. The maximum difference between the two values at positions 180° apart reaches 749nT, indicating that the sensor's leveling with the bubbles does not accurately represent the perpendicularity of the Z magnetic axis, suggesting the presence of directional error. We continued adjusting the three base screws to make the Z element's output values as close as possible at different positions. Data from the 2nd to the 6th sets represent the stepwise adjustment of the Z component output values by the base screws. The data from the 6th set are the measurement results when the base screws are adjusted to their limits. At this time, the difference between the two values of the Z component at positions $180°$ and $90°$ apart is 1nT and 22nT, respectively. Since the measurement experiment was not conducted in a uniform magnetic field laboratory, even during a quiet magnetic period, there will still be small diurnal variations, making it difficult to achieve an ideal state where the Z component remains unchanged at any position when

rotating the platform. There is a certain error. The maximum difference between the Z component values at different positions is 22nT. If this value is projected onto the magnetic declination D direction, compared to the output value range of -50-50nT when the magnetic declination D is oriented, its impact is small. Therefore, we believe that the Z magnetic axis is now in a vertical position. Obviously, the bubbles of the instrument itself are no longer centered, and we level a pair of external levels previously fixed on top of the sensor.

Subsequently, we measured the offset of the D magnetic axis according to the method shown in Figure 2. By rotating the platform, we read the output values of the D element in two directions 180° apart, and calculated the D magnetic axis offset to be 109nT. Using the formula to convert the D magnetic axis offset from nT to angle, it is expressed as: $\theta=\arcsin\frac{S_0}{H}$ . In the formula, H takes the annual average value of the H component. The known H value of Hongshan Station is 29,600 nT, and the $\theta$ value can be calculated to be approximately 0.2°.

Finally, the instrument under test was moved to the relative recording room of the Hongshan observatory. We first checked and leveled the external bubbles fixed at the top of the sensor to correct the Z magnetic axis directional error. Then, we adjusted the sensor to ensure the output value of the D element to be (109±50) nT, completing the correction of the D magnetic axis directional error, and initiating the instrument to begin recording observations.

**Table 1 Adjustment Results of the FGM-01 Instrument for the Verticality of the Z Magnetic Axis**

| Level Angle | Z-Element Output Value(nT) | | | | | |
|---|---|---|---|---|---|---|
| | 1 | 2 | 3 | 4 | 5 | 6 |
| 290° | 215 | 198 | 85 | 155 | -150 | -158 |
| 200° | -59 | -131 | -177 | -115 | -134 | -135 |
| 110° | -534 | -518 | -403 | -161 | -160 | -157 |
| 20° | -215 | -146 | -94 | -127 | -142 | -136 |

It should be noted that we know the daily variation of the geomagnetic field is very small. When conducting experiments, it is advisable to choose a period when the magnetic field is calm and the temperature is stable. It can be considered that at this time, the geomagnetic field is stable and uniform.

**4 Analysis of Results**

To verify the correction effects mentioned above, the data from the instrument before and after calibration was compared in the following two ways.

**4.1 Comparative analysis of calibration results for daily variation records accuracy**

The purpose of the calibration for daily variation records accuracy is to examine the accuracy of the fluxgate magnetometer in recording diurnal variation data. The specific method is as follows: On a selected day, an absolute measurement is carried out every hour, and two sets of valid data are obtained for each measurement. The precision and stability are measured by calculating the change in the baseline value and the standard deviation (Gao et al.,1991; Zhang and Yang,2011). The tested instrument underwent diurnal variation calibration both before and after correction, with 10 absolute measurements each time, as shown in Figure 5. As can be seen from Figure 5, there is a clear diurnal variation pattern on the baseline value curves of the tested instrument before correction, with the maximum baseline value change $D_B$ being 0.26′, $H_B$ being 1.7nT, and $Z_B$ being 1.0nT, and the standard deviation for $D_B$ being 0.07′, $H_B$ being 0.5nT, and $Z_B$ being 0.3nT. After correction, the maximum baseline value changes for each element were $D_B$ being 0.07′, $H_B$ being 1.0nT, and $Z_B$ being 0.6nT, with standard deviations of $D_B$ being 0.02′, $H_B$ being 0.3nT, and $Z_B$ being 0.2nT. Compared to the pre-correction data, there was a reduction in both the maximum baseline value changes and standard deviations for each element, with the D element showing the most significant decrease in maximum baseline value change, by 0.19′. The results indicate that the observational data accuracy of the tested instrument post-correction is significantly superior to that of the pre-correction, and it more truly reflects the diurnal variation of the geomagnetic field.

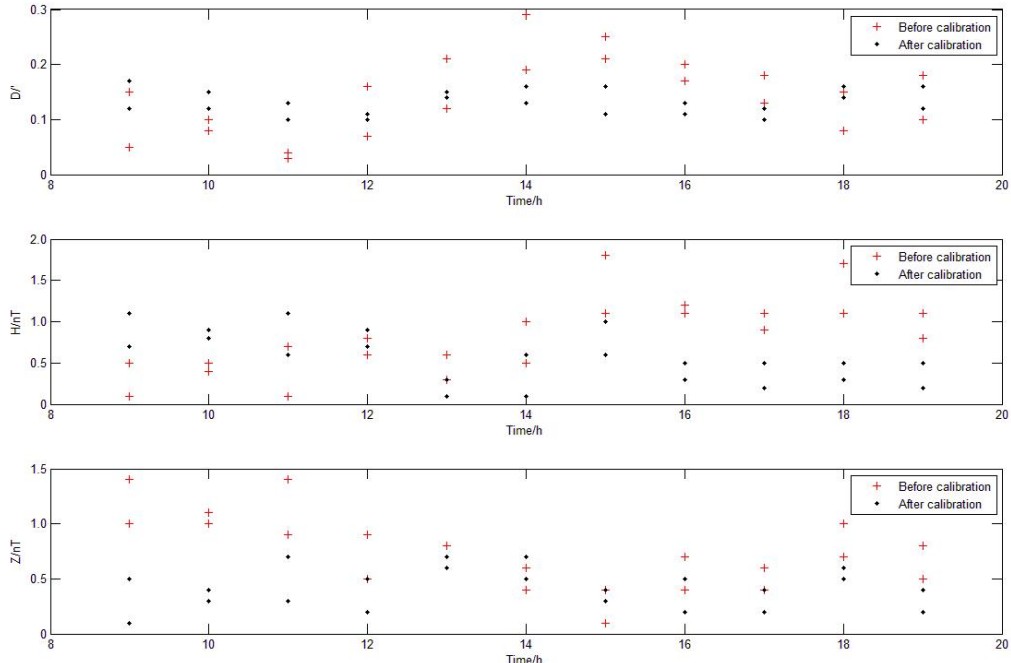

**Figure 5: Baseline values obtained through daily calibration before and after calibration of the test instrument**

**4.2   The comparison of the difference curve of daily variation before and after instrument calibration**

The comparison of the difference curve of daily variation can generally describe the consistency of data from different instruments at a station. Using the standard instrument GM4 located in the relative recording room of Hongshan observatory as the reference instrument, the minute values of the instrument under test are compared with those of the reference instrument. The difference curves before and after correction are shown in Figure 6, where 6(a) and 6(b) respectively represent the difference curves for geomagnetic quiet days and geomagnetic disturbed days. It can be observed that the difference between the instrument under test and the reference instrument is significant before calibration, especially for the D component. Even when the magnetic field is relatively stable, the maximum variation in the difference of the D component still reaches 0.22′. After calibration, the differences in magnetic field components between the instrument under test and the reference instrument are significantly reduced, particularly for the D and Z components, with the difference curves coming much closer to a straight line compared to the reference instrument.

Select the observation data of the instrument before calibration (May 2022) and after calibration (May 2023) for comparison. From these data, respectively select five days of magnetically quiet days and five days of magnetically disturbed days, and calculate the range of relative difference amplitude with the reference instrument and its average range (Table 2). As can be seen from Table 2, the average change range of the relative difference amplitude of the D and Z components before calibration is "-0.11 ~ 0.13' and -0.4 ~ 0.6nT", with the average change range of the amplitude being similar on magnetically quiet days and magnetically disturbed days. The average change range of the amplitude of the H component is almost twice as large on magnetically disturbed days compared to magnetically quiet days. After calibration, the H component shows a slight decrease compared to before, and the improvement effect of the D and Z components is very significant. The average change range of the amplitude on magnetically quiet days is "-0.02 ~ 0.03' and -0.2 ~ 0.3nT", and on magnetically disturbed days it is "-0.05 ~ 0.04' and -0.3 ~ 0.2nT", with the average change range of the amplitude being significantly reduced compared to before calibration. This indicates that the above-mentioned orientation method has a good calibration effect on the magnetic field components.

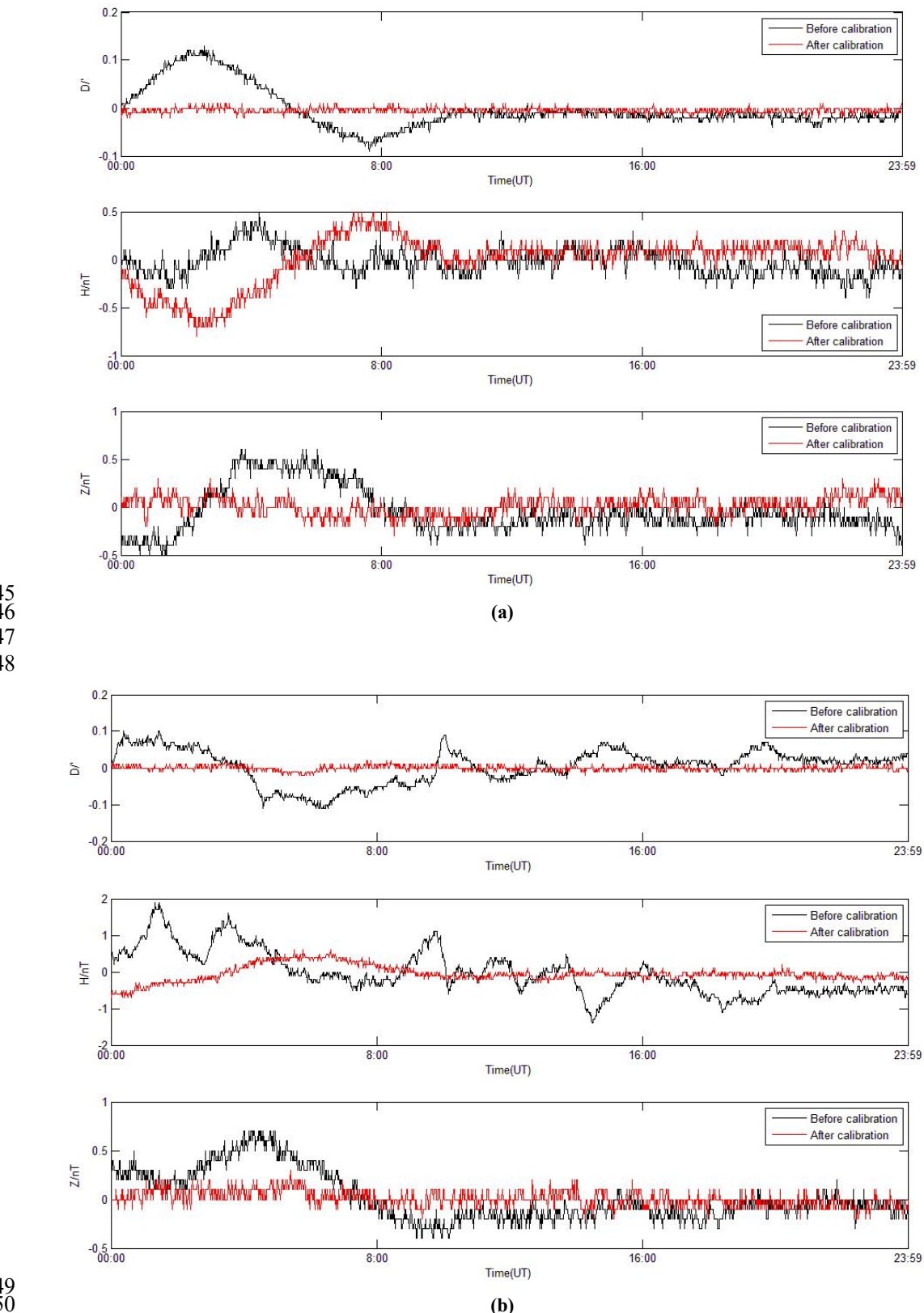


(a)


(b)

Figure 6: The difference curve between the tested instrument before and after calibration and the reference
instrument
(a)geomagnetic quiet day (b)geomagnetic disturbance day
**Table 2 The Range of Relative Difference Amplitudes Between the Test Instrument and the Reference**
**Instrument**

| | | before calibration | | | | after calibration | | |
|---|---|---|---|---|---|---|---|---|
| | Date | D (′) | H (nT) | Z (nT) | Date | D | H | Z |
| Magnetic Quiet Day | 1 May | -0.11 ~ 0.12 | -0.7 ~ 1.0 | -0.4 ~ 0.6 | 5 May | -0.02 ~ 0.03 | -0.7 ~ 0.6 | -0.3 ~ 0.2 |
| | 19 May | -0.12 ~ 0.13 | -0.4 ~ 0.8 | -0.3 ~ 0.8 | 8 May | -0.02 ~ 0.02 | -0.5 ~ 0.4 | -0.2 ~ 0.2 |
| | 20 May | -0.11 ~ 0.16 | -0.5 ~ 0.4 | -0.4 ~ 0.4 | 19 May | -0.03 ~ 0.03 | -0.5 ~ 0.4 | -0.2 ~ 0.3 |
| | 21 May | -0.11 ~ 0.12 | -0.5 ~ 0.3 | -0.5 ~ 0.6 | 21 May | -0.01 ~ 0.03 | -0.4 ~ 0.4 | -0.3 ~ 0.3 |
| | 25 May | -0.09 ~ 0.14 | -0.7 ~ 0.6 | -0.4 ~ 0.7 | 25 May | -0.02 ~ 0.02 | -0.6 ~ 0.6 | -0.2 ~ 0.3 |
| | Mean | -0.11 ~ 0.13 | -0.6 ~ 0.6 | -0.4 ~ 0.6 | Mean | -0.02 ~ 0.03 | -0.5 ~ 0.5 | -0.2 ~ 0.3 |
| Magnetic Disturbed Day | 5 May | -0.11 ~ 0.13 | -1.4 ~ 0.4 | -0.3 ~ 0.9 | 1 May | -0.02 ~ 0.04 | -0.8 ~ 0.9 | -0.3 ~ 0.3 |
| | 6 May | -0.12 ~ 0.09 | -1.5 ~ 1.6 | -0.5 ~ 0.5 | 11 May | -0.07 ~ 0.03 | -1.3 ~ 1.0 | -0.3 ~ 0.2 |
| | 8 May | -0.11 ~ 0.13 | -0.8 ~ 1.2 | -0.4 ~ 0.7 | 14 May | -0.07 ~ 0.05 | -0.9 ~ 1.4 | -0.2 ~ 0.3 |
| | 17 May | -0.13 ~ 0.16 | -1.7 ~ 1.2 | -0.6 ~ 0.6 | 15 May | -0.05 ~ 0.02 | -1.0 ~ 0.8 | -0.2 ~ 0.3 |
| | 31 May | -0.11 ~ 0.17 | -1.3 ~ 1.0 | -0.4 ~ 0.5 | 27 May | -0.03 ~ 0.05 | -1.0 ~ 0.9 | -0.3 ~ 0.1 |
| | Mean | -0.12 ~ 0.14 | -1.3 ~ 1.1 | -0.4 ~ 0.6 | Mean | -0.05 ~ 0.04 | -1.0 ~ 1.0 | -0.3 ~ 0.2 |

**5 Discussion and Conclusion**

This paper has analyzed the generation mechanism of orientation errors in triaxial fluxgate magnetometers and conducted a station experiment on an FGM-01 instrument using a self-made measurement device. The experimental and research results show that orientation errors occur in both the Z and D magnetic axes of the sensor, and it is necessary to correct these errors. The observational data, after correction for orientation errors, demonstrated a significant reduction in both the standard deviation of baseline values and the amplitude of differences when compared to a reference instrument, proving the effectiveness of the correction method. The measurement device used in the experiment is low-cost, simple to operate, and easy to disseminate, boasting a high performance-to-price ratio. In this study, the author found that the improvement effects on the D and Z components are more pronounced, whether on magnetically quiet or disturbed days, but not as significant for the H component. This indicates that the accuracy of geomagnetic daily variation records is influenced by factors other than orientation errors, including orthogonality, among others. We will continue to examine the impact of instrument orthogonality and correction methods in future work. The research presented in this paper provides a reference for the standardized installation and regular adjustment of orientation in fluxgate magnetometers at geomagnetic stations.

**Acknowledgments**

The geomagnetic observation data in this article are sourced from the National Geomagnetic Network Center of the Institute of Geophysics, China Earthquake Administration. Senior Engineer Li Xijing from the Qianling Station in Shanxi Province proposed the measurement method for directional errors and guided the specific experimental procedures. We would like to express our gratitude here as well.

**Funding Information**

Supported by Science for Earthquake Resilience of China (XH23006A) and Special Project of Hongshan Scientific Observation of China (DZ2021110500003) .

**Competing Interests**

The authors have no competing interests to declare.

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
