# Peer review of "Analysis of Orientation Errors in Triaxial Fluxgate"

_Geoscientific Instrumentation, Methods and Data Systems, 2024_

## Referee Comment (RC2)

"General comments"

The paper " Analysis of Orientation Errors in Triaxial Fluxgate Sensors and Research on Their Calibration Methods" by Xiujuan Hu et al. is a contribution to calibration of the fluxgate magnetometers for geophysical observatories. The scientific and technical questions addressed in the paper are within the scope of GI.

The paper proposes two methods of accurate orientation of the observatory three-axis magnetic sensor and their practical implementation is described. The performance of the calibrating procedures is evaluated by analyzing the scatter of the magnetometer's baseline values before and after the tests, as well as by comparing its difference signals relatively a reference variometer.
The first method exploits sensor rotation around a nearly vertical line, and the leveling of the sensor head until the Z component deviations in four positions are as small as possible.
This is not a new idea, for instance, Jankowski and Sucksdorff in 1996 proposed in the first paragraph of the subsection 8.4 of their book "Guide For Magnetic Measurements And Observatory Practice" the following procedure:
"It is possible to make all the determinations also in the following simple way: A turntable is needed for the sensor assembly. This can be an old theodolite. First, level the turntable carefully so that its axis is vertical. Then place the sensor assembly on the turntable (controlling the leveling of the turntable) so that the vertical sensor is vertical with high accuracy. The sensor assembly has to be adjusted until turning the table does not change the Z-reading. We now know that the Z-sensor is vertical and
measures only the Z-component. The levels of the sensor now have to be adjusted so that when installing the sensor at a new place it takes up the same vertical position."

The second method solves the problem of how to orient the horizontal sensor D (or Y) perpendicular to the magnetic meridian. This means that the projection of the horizontal component of the Earth's magnetic field onto the axis of the selected sensor is zero and its output signal should also be zero, ideally. Due to the offset of the D (or Y) sensor, there arises an angular error. This offset is the sum of the sensor zero offset and the projection of the vertical component of the Earth's magnetic field due to the slight deviation of the D (or Y) magnetic axis from the horizontal plane.
The authors (rotating the sensor around the vertical line) measure two outputs of D-sensor in two opposite directions and calculate the offset of this sensor. Jankowski and Suksdorf described exactly the same procedure in their book mentioned above (from the second to the fifth paragraph of subsection 8.4). Furthermore, subsection 8.4 describes a method for determining the non-orthogonality of the X, Y horizontal sensors using the same turntable setup.
In my opinion, the ideas and concepts of calibrating the orientation error of three-axis fluxgate sensors proposed by the authors are an incomplete repetition of the known approaches used in the practice of geomagnetic observatories.

The authors based their research on analysis of a large number of related studies and this is reflected in the list of references. Unfortunately, the book "Guide For Magnetic Measurements And Observatory Practice" by Jankowski and Sucksdorff was not included into this list.

The title and abstract clearly and completely represent the contents of the paper.
The overall presentation of the study results is well structured, but not clear enough without adding more information. Abbreviations, symbols and units are fairly defined and used.

"Specific comments"

**page 3, lines 116-117**
*"The measurement part of this experiment was conducted in the absolute measurement room of Hongshan observatory."*
What are the elements of the Earth's magnetic field vector at the calibration site? At least the horizontal intensity H must be known in order to convert the values of D and Z (presented in nT in Section 3) into angular errors.

**page 3, lines 119-121 and Figure 3(c)**
"*After leveling the non-magnetic theodolite, we began adjusting the sensor's base angle screws to center two mutually perpendicular bubbles.*"
How sensitive are the bubble levels glued to the top of the sensor under test? How stable is the position of the bubble levels relatively the Z magnetic axis?

**page 4, lines 128-130 and Table 1 (the 6th set of data)**
"*The 6th set of data reflects the measurement result when the base angle screws were adjusted to their extreme positions, with a difference of only 1nT between the two values at positions 180°apart...*"
Why are the Z-sensor output values not equal at all four positions (290°, 200°, 110°, 20°)? What is the reason for the 22 nT difference between the values in the (290°, 110°) and (200°, 20°) position groups?
Does this mean that the magnetic field at the calibration site is non-uniform? If so, how does this non-uniformity affect the offset measurements of the D-sensor?

"Technical corrections"

**page 3, Figure 3**
The captions for Figure 3 b and c are mixed up. (These typos had been corrected in the manuscript version uploaded 2024.07.05)

**page 5, lines 177-179**
*"The difference curves before and after correction are shown in Figure 5.geomagnetic quiet day and geomagnetic disturbance day are shown in Figure 5(a) and 5(b)."*
Does it have to be two sentences?

**page 5, lines 185-187 and Table 2**
*"Selecting data from five geomagnetically quiet days and five disturbed days before and after the calibration of the instrument under test, we computed the range of difference with the reference instrument and calculated the average amplitude (Table 2)."*
It is not clear how the average amplitude can be a negative number. In my opinion, the table columns "D", "H", "Z" contain the range of the difference signal, but not its average amplitude. Probably, the table title has to be something like that "The range of the daily variation difference..."
What do the numbers in the columns of the "Date" table mean? For instance, the numbers 5.1, 5.5, 5.8 are presented in the both parts of the table. Does that mean those days were both magnetic quiet and disturbed at the same time?
How can the date in the "before calibration" column be greater than the date in the "after calibration" column? For example, 5.19 – 5.8, 5.20 – 5.19, 5.5 – 5.1, 5.17 – 5.15, 5.31 – 5.27.

**page 8, line 249**
Wrong link: https://doi.org/10.11939/jass.2016.01.013

**page 8, line 257**
Wrong link: https://doi:10.6038/pg2019CC0024
Must to be: https://doi.org/10.6038/pg2019CC0024 ?

**page 8, line 274**
Wrong link: https:/doi.10.11939/jass.2017.03.012

---

## Author Response (AR1)

Dear Editor,

We have two revised the article based on the reviewers' comments and have responded to the relevant comments in the comments section.

The first modification is as follows:

1. The main focus of this paper is the analysis and experimentation of errors generated during the directional installation of instruments, and it does not involve the analysis of instrument orthogonality. During the experiments, it is assumed that the three axes of the instruments are in an ideal state of complete orthogonality. Due to the lack of detailed descriptions in the original text, there is a tendency to confuse the concepts of directionality and orthogonality. We have now added relevant explanations about the directional errors and principles of the instruments in the text and have removed the confusing statements to make the content more detailed and complete, and the expression more accurate.

2. The original text lacked an introduction to the non-magnetic rotating platform and a description of the platform's state during the experiment. Since this platform can achieve absolute level by adjusting the theodolite dial, and the measurement data are conducted based on the horizontal plane, adjusting the base screw of the instrument so that the Z-axis output values are close at different positions indicates that the Z-axis is perpendicular to the platform and therefore in the direction of geographic vertical. The original text has been supplemented with relevant descriptions.

The second modification is as follows:

1. The article cites "Guide for Magnetic Measurements and Observatory Practice" by Jankowski and Sucksdorff. See Page 4 , Line 142-144 and Page 9 , Line 297-299.

2. After calculating the D magnetic axis offset, it can be converted from nT to degrees using a formula. See Page 5 , Line 176-179.

3. Descriptions of the accuracy and installation positions of the two additional water bubbles on the sensor have been added. See Page 4 , Line 135-141.

4. During the verticality adjustment of the Z magnetic axis, an explanation was given for the result of the Z output values differing by 22nT at positions 180° apart. See Page 4,5 , Line 162-173.

5. The textual descriptions between Figure 5, Figure 5(a), and Figure 5(b) have been revised. See Page 6, Line 220-222.

6. The selection times for five quiet days and five disturbed days before and after calibration have been rephrased. See Page 6, Line 229-242. The header of Table 2 has been revised. See Page 7, Line 254-255.

7. One reference with an incorrect format has been revised. See Page 9, Line 304-305. The DOIs of two other references have been verified; the data is correct, but they cannot be accessed for unknown reasons. Should the DOI data be deleted or retained? See Page 9, Line 294-296. See Page 9, Line 320-322.

Best regards,

Xiujuan Hu